# Impact of systemic anticancer therapy in pediatric optic pathway glioma on visual function: A systematic review

**Carlien A. M. Bennebroek**[1]*, **Laura. E. Wijninga**[1], **Jaqueline Limpens**[2], **Antoinette Y. N. Schouten-van Meeteren**[3], **Peerooz Saeed**[1]

**1** Department of Ophthalmology, Cancer Center Amsterdam, Amsterdam UMC, University of Amsterdam, Amsterdam, The Netherlands, **2** Medical Library, Amsterdam UMC, University of Amsterdam, Amsterdam, The Netherlands, **3** Department of Neuro-Oncology, Princess Máxima Center for Pediatric Oncology, Utrecht, The Netherlands

☯ These authors contributed equally to this work.
* c.a.bennebroek@amsterdamumc.nl

**Data Availability Statement:** All data are available within the systematic review.

**Funding:** The authors received no specific funding for this work.

## Abstract

Pediatric optic pathway glioma (OPG) can seriously decrease visual function in the case of progression. Systemic anticancer therapy (SAT) is considered the treatment of first choice for unresectable OPG. New SAT modalities for the treatment of progressive OPG have been introduced in the last decade, including VEGF and MAPK pathway inhibition. This systematic review evaluated the effect of SAT on change in visual acuity and visual field in OPG. A systematic review was performed on SAT for OPG (January 1990 to August 2020). MEDLINE and EMBASE (Ovid) were searched for studies reporting on change in visual acuity and visual field after treatment with SAT for OPG. Overall, 11 series, including 358 patients, fulfilled the eligibility criteria. After follow-up of median 3.7 years (range: cessation of SAT– 8.2 years), improvement in binocular VA was found in 0–45% of studies, stability in 18–77% and a decrease in 0–82%. Two studies reported on change in visual field (improvement in 19% and 71% of patients), although either the change was not defined or the testing strategy was lacking. Considerable heterogeneity was present among the included studies, such as variety in the combinations of SAT administered, status of neurofibromatosis type 1, definition regarding change in visual acuity, 1- or 2-eye analysis, diversity in anatomic location, and extent of follow-up, all of which made meta-analysis inappropriate. This systematic review suggests that the impact of SAT in OPG on visual function is still unclear. The wide ranges reported on the efficacy of SAT and the observed heterogeneity highlight the need for prospective studies with uniform definitions of outcome parameters.

## Introduction

Optic pathway glioma (OPG) is considered a rare subtype of pediatric low-grade glioma (LGG) located in the optic pathway from the optic nerve to the optic tract. OPG presents on average at the age of 3–9 years (range 0–7 years), either in association with neurofibromatosis

**Competing interests:** The authors have declared that no competing interests exist.

**Abbreviations:** CEBM, Centre for Evidence-Based Medicine; CT, Chemotherapy; ICO, International Council of Ophthalmology; JBI, CA-Joanna Briggs Institute–Critical Appraisal Tool; LGG, Low-grade glioma; LogMAR, Logarithm of the minimum angle of resolution; MRI, Magnetic resonance imaging; MAPK, Mitogen-Activated Protein Kinase; (M)DC, (Modified) Dodge Classification; NF1, Neurofibromatosis type 1; nNF1, No association with neurofibromatosis type 1; OCT, Optical coherence tomography; OPG, Optic pathway glioma; OS, Overall survival; PFS, Progression-free survival; RNFL, Retinal nerve fiber layers; SAT, Systemic anticancer therapy; VA, Visual acuity; VEGF, Vascular endothelial growth factor; VF, Visual field; VD, Ventricular drainage.

type 1 (NF1) (incidence: 10–60%) or without NF1 (nNF1) [1–3]. Most frequently, OPG is a pilocytic astrocytoma followed by a pilomyxoid astrocytoma. Treatment is indicated in the case of radiologic or clinical progression, including significant visual deterioration or neurological symptoms, as OPG may remain stable in volume (presumed to be mostly NF1) or, rarely, regresses spontaneously in the case of NF1 OPG [4].

Regardless of the high rate of 5-year overall survival (OS) after treatment (89–95%) [1, 5–7], loss of visual function can be extensive and may have a significant impact on the quality of life [8]. Long-term analysis of visual outcome after diverse treatment strategies for OPG presents binocular BCVA ≤ 20/200 in 16–26% [9, 10] or < 20/100 in 58% of patients [11].

Systemic anticancer treatment (SAT), mostly chemotherapy, is considered the treatment of first choice for OPG because the possibility of surgery is often limited or not feasible due to the risk of damaging visual, neurologic or endocrine function [12]. As 45–66% of OPG progresses during or after first-line SAT [1, 6], successive systemic treatment is often necessary. In non-surgical cases, second-line up to fifth-line SAT can be applied when there is progression. Maximum delay in the application of radiotherapy is highly preferred because of its long-term side effects, considering endocrine deficiencies, vasculopathy, and neurocognitive impairment [13–15]. Many different first-line or next-line SAT regimens have been introduced since the first results were published in 1976 [1, 16–19], with increased frequency of introduction from 1990 onwards. Initial therapy is frequently carboplatin-based. In Europe, treatment with carboplatin and vincristine over a period of 18 months represents the current first-line strategy proposed by the Societé Internationale d'Oncologie Pédiatrique (SIOP) [1]. The anti-vascular endothelial growth factor (anti-VEGF) agent bevacizumab was introduced in 2009 as the next-line treatment for progressive OPG [20], as angiogenesis plays a vital role in the growth of LGG. Results on treatment outcome show a rapid radiological response [20, 21] with anecdotally profound visual improvement [22, 23]. Internationally, bevacizumab is not part of the standard of care for progressive pediatric OPG, as recurrence of progression after discontinuation of bevacizumab is frequent (15–93% within 6 months after cessation) [21, 23–25] and toxicity profiles are still being studied. Recently, the effect of targeted inhibition of the MAPK pathways on low-grade glioma is being increasingly studied regarding dose, treatment duration, effectiveness and toxicity [26–28].

Decrease of visual function is mostly more prominent than neurologic dysfunction and is one of the leading clinical indicators for starting treatment for OPG [29]. After treatment, decreased visual function is also considered the main invalidating outcome parameter. Currently, treatment evaluation is based on a combination of radiological response and clinical evaluation. Regarding the latter, to date, visual acuity (VA) is the only visual outcome parameter that can represent change in function after treatment [30]. Visual field is assumed to mirror VA function [30], but there is insufficient evidence to substantiate this assumption.

So far no correlation has been found between (the current 2-dimensional) radiological response after SAT and change in visual function (analysed mainly with NF1 patients) [29–31]; hence the focus on the clinical effect of therapy on visual function is essential.

In 2010, Moreno et al. [32] published a systematic review to evaluate the effect of SAT on VA, which suggested several trends: 14% improvement in VA, 47% improvement of stability of VA and a decrease in VA of 39% of patients after chemotherapy. The way in which change in VA was defined was not evaluated. No statistical analysis could be performed due to the heterogeneity of the included studies. The authors concluded by urging for standardisation of treatment indications and evaluation.

In 1997 the NF1 OPG Task Force consensus statement provided rational guidelines for the diagnosis and treatment of OPG in NF1 [33]. Updates in 2007 and 2017 added a focus on visual function represented by VA measurement, which included the proposed usage of

validated test modalities to measure the VA suitable per age category, and definitions of an age-based norm for normal VA [34–36]. In 2020 the RAPNO working group suggested defining *change in VA* by change in $<> 0.2$ LogMAR [37], substantiated by the application of this definition in prior studies [29, 30]. In view of the fact that recommendations have been developed and new treatment modalities have become available, we performed a systematic review to evaluate the effect of SAT on visual function (VA and VF), including VEGF and MAPK inhibition.

## Materials and methods

### Search strategy

This study was conducted in accordance with the PRISMA guidelines (PRISMA checklist October 2015) and registered in the PROSPERO international prospective register of systematic reviews (Reg. no. CRD42020125576, see S1 File).

A medical information specialist (JL) performed a comprehensive search of OVID MEDLINE (using the PubMed interface) and OVID EMBASE from January 1990 until August 5, 2020. Empirically we found that we would miss relevant studies which did not mention specific drugs or mentioned only OPG treatment. Therefore we constructed a search consisting of three parts combined with a pediatric search filter: (1) OPG + chemotherapy, (2) OPG + other anticancer agents, and (3) cohort studies on managing pediatric OPG (major topic). Conference abstracts were excluded from EMBASE. Detailed searches for both databases are available (see S1 and S2 Tables). Reference lists and articles cited in the included papers, as well as relevant reviews, were crosschecked for additional relevant studies.

Titles, abstracts and full-text articles were screened independently by two authors (CB and LW). Differences in opinion were resolved through discussion; if necessary, a third author (PS) was consulted.

### Eligibility criteria

The primary endpoint of this review was the percentage and range of OPG patients with change in VA after SAT, divided into 3 categories: improvement, stability, or decrease of VA. The secondary endpoint was change in VF.

Studies were included when or if they (1) reported on change of visual acuity in children ($\leq$ 18 years) after receiving SAT for OPG; (2) included a minimum of 10 patients per study; (3) were written in any language, as long as the original authors were willing to translate their manuscript into English; and (4) reported on patients with or without surgical treatment (biopsy/ventricular drainage (VD)/tumor resection) prior to SAT. Studies containing the results of patients who required additional therapy after SAT were also included.

Studies reporting on radiotherapy prior to SAT were excluded. When the results of the studies appeared to overlap, the study with the most recent data was included.

### Data collection

The following data were extracted: study characteristics, patient characteristics, variables regarding visual function, and prognostic factors for a decrease in VA and/or VF (Tables 1 and 2).

### Critical appraisal

Assessment of methodological quality was performed in parallel by two authors (CB, PS). Study quality was weighed using the Oxford Centre for Evidence-Based Medicine (CEBM) evidence rating system. If studies were case series, the Joanna Briggs Institute Critical Appraisal

**Table 1. Study & patient characteristics.**

| Author (year) | Country | Study design | Sample size change in VA (total population per study) | NF1 nNF1, registered in total study population | Age started CT: mean/ median (SD/range) (years) | Time interval (median/range) VA started CT– final VA (years) | (M)DC stage | Definition of change in VA | VA ↑ binoc: N (%) | VA ↔: binoc: N (%) | VA ↓: binoc: N(%) |
|---|---|---|---|---|---|---|---|---|---|---|---|
| Massimino et al. (2002) [17] | Italy | Prospective, multicenter | 22 (34) | 8/ 26[a] | 3.2 (0.3–15.6) | 3.7 (0.8–10) | 2 | No definition | 10 (45) | 7 (32) | 5 (23) |
| Dalla Via et al. (2007) [42] | Italy | Prospective, monocenter | 11 (20) | 20 NF1 | 2.2 (1.1–4.2)[b] | 6.3 (0.4–18)[c] | 1, 2, 3 | No definition | 0 (0) | 2 (18) | 9 (82) |
| Massimino et al (2010) [43] | Italy | Prospective, center: NEP | 17 (37) | 7/ 30[a] | 6.0 (0.5–16.5)[a] | End of CT cycle | 1, 3 | No definition | 7 (41) | 10 (59) | 0 (0) |
| Shofty et al. (2011) [45] | Israel | Retrospective, multicenter | 19 (19) | 11/ 8 | 5.2 (1–9.4) | 4.2 (0.3–11) | 2, 3, 4 | No definition | 1 (5) | 4 (21) | 14 (74) |
| Fisher et al. (2012) [40] | USA, UK, Australia, Canada | Retrospective, multicenter | 88 (115) | 88 NF1 | 4.0 (0.5–16.2) | 3 months after completion of CT cycle | 1, 2, 3, 4 | ≤ = ≥ 0.2 Snellen lines | 28 (32) | 35 (40) | 25 (28) |
| Kalin-Hadju et al. (2014) [31] | Canada | Retrospective, monocenter | 14 (17) | 10/ 7[a] | 3.4 (3.2–6.6) | 8.2 (3.5–12.9) | 1, 2, 3, 4 | Change per (WHO) category of CVI scale | 1 (7) | 7 (50) | 6 (43) |
| Dodgshun et al. (2015) [9] | Australia | Retrospective, monocenter | 35 (104) | 33/ 71[a] | 4.6 (0.4–12) | At cessation of CT/ (6.5 (ND)[D] | 1, 2, 3, 4 | Change in ICO category (13) | 5 (14) | 27 (77) | 3 (9) |
| Prada et al. (2015) [44] | USA | Retrospective, monocenter | 22 (826) | 22 NF1 | 4 (1.8–12) | Not registered | 1, 2, 3, 4 | No definition | 4 (17) | 6 (27) | 12 (56) |
| Doganis et al. (2016) [36] | Greece | Retrospective, monocenter | 16 (20) | 15/5[a] | 5.3 (1.5–11.4) | End of CT cycle/ 5.2 (2.8–9.4)[D] | 1, 2, 3 | No definition | 7 (44) | 7 (44) | 2 (12) |
| Lassaletta et al. (2016) [18] | Canada | Prospective, multicenter | 24 (54) | 13/ 41[a] | 8 (0.7–17.2) | 5 yr (ND) | 2, 3, 4 | No definition | 5 (21) | 15 (62) | 4 (17) |
| Falzon et al. (2018) [29] | UK | Prospective, multicenter | 90 (90) | 46/ 44 | 3.8 (0.8–14) NF1, 3.2 (0.4–15) nNF1 | 6.5 (2.0–10.2) | 1, 2, 3 | ≤ = ≥ 0.2 LogMAR | 19 (21) | 35 (39) | 36 (40) |
| **Total** | | | 358 (1,336) | 178/52 | | | | | | | |

Abbreviations: CVI: Childhood Visual Impairment; CT: chemotherapy; FU: follow-up; ND: no data; NEP: no extraction possible; NF1: neurofibromatosis type 1; nNF1: no systemic association with neurofibromatosis type 1; (M)DC: (Modified) Dodge Classification; SD: standard deviation; TX: treatment, VA: visual acuity.

*: monocular

[a]: results only available from total population of study

[b]: age at diagnosis, age at start of tx not available

[c]: interval age at diagnosis–final VA

[D]: long-term data available, change in VA not registered in this table. See Table 3.

(JBI-CA) tool for case series "Checklist for Case Series" [38] was used, in which bias is evaluated by means of 10 questions answered by Yes, No or Unsure. As several of the included studies did not focus primarily on the effect of SAT, we performed a critical appraisal pertinent to the primary endpoint of this review. If no statistical analysis was performed, question 10 was evaluated as Unsure. We considered a low risk of bias if the Yes answers were ≥ 50%, a high risk of bias if the No answers were ≥ 50%, and uncertain risk of bias if the Unclear answers were ≥ 50%.

**Table 2. SAT combinations, prior surgical intervention, VA test, VF test and prognostic factors on decrease of VA.**

| Author (year) | SAT (N) | Previous TX (N) | Visual tools | VF: type of test | Change in VF after SAT | Prognostic factors on decrease of VA |
|---|---|---|---|---|---|---|
| **Massimino et al. (2002)** [17] | Cispl—ETO (34)[b] | ND | ND | - | ND | ND |
| **Dalla Via et al. (2007)** [41] | CB-VC | No | TAC, LH, Snellen | ND | ND | ND |
| **Massimino et al. (2010)** [42] | Cispl- ETO (37)[A] | Prior CT/ RT: no, SX: ND | ND | - | ND | ND |
| **Shofty et al. (2011)** [43] | CB- VC (19) | No | Snellen, self-made grading system | - | ND | ND |
| **Fisher et al. (2012)** [30] | CB-VC (105), CB (9), VB (1) | Biopsy + (ND), SX: ND, prior CT/ RT: excluded | TAC, Lea, HOTV, Snellen | ND | Total population: 5 (19%) improved, 10 (38%) remained stable, 11 (42%) decreased. No definition of change | Location in optic tracts/radiation: (OR 3.0; 95% CI: 1.1–8.3; P = 0.032) |
| **Kalin-Hadju et al. (2014)** [31] | CB-VC (7), CB (4), CB-VB (2), CP (2), TPCV (1), VC-AC (1) | 3/17 VD, prior CT/RT: ND | BFP, TAC, Allen card, Snellen | - | ND | ND |
| **Dodgshun et al. (2015)** [9] | CB (38), CB-VC (4 )[B] | Biopsy + (ND), SX/ RT: ND | Snellen, Kay Pictures | ND | 7/35 (20%) abnormalities at diagnosis: 2/ 7 (29%) improved, 5/7 (71%) stable (no definition of change) | ND |
| **Prada et al. (2015)** [44] | CB/VC (21), VC/DC (1)[B] | ND | ND | - | ND | ND |
| **Doganis et al. (2016)** [45] | CB-VC (16), of which switch to VB (5) due to allergy | ND | Snellen, Kay Pictures | - | ND | ND |
| **Lassaletta et al. (2016)**[18] | VB (24) | Previous sx/ biopsy + (ND) | ND | - | ND | ND |
| **Falzon et al. (2018)** [29] | CB-VC (46), CB & VC or ETO (44) | Biopsy + (ND), S: ND, prior CT/RT: excluded | TAC, Snellen | - | ND | -NF1: Age ≤ 5 years (OR 5.3; 95% CI: 1.0–26.7; P = 0.04) and (M)DC stage 3 (OR 7.1; 95% CI: 1.8–33.3; P = 0.006) -nNF1: No prognostic factor found |

Abbreviations: AC; actinomycin, BFP: binocular fixation preference; CB: carboplatin; CI: confidence interval; Cispl: cisplatin; DC: dactinomycin; ETO: etoposide; HOTV: HOTV eye test chart; (M)DC: (Modified) Dodge Classification; ND: no data; NF1: neurofibromatosis type 1; nNF1 No systemic association with neurofibromatosis type 1; P: P-value; PCZ: procarbazine; OR: odds ratio; RT: radiotherapy; SAT: Systemic Antitumor Therapy; SX: surgery; TAC: Teller Acuity Cards; TPCV: thioguanine, procarbazine, lomustine and vincristine; TX: treatment; VA: visual acuity; VB: vinblastine; VC: vincristine; VD: ventricular drain; VF: visual field

[A]: Results only available for total population that received SAT.

[B]: Subpopulation that received SAT

## Statistical analysis

Study characteristics, patient characteristics and definition of change in VA and VF between the start of and after treatment with SAT were reported descriptively. Data regarding change in VA were reported as range (percentage) and cumulative proportion (number and percentage of change) (see Discussion) or are calculated data.

## Results

### Search

The search strategy identified a total of 818 studies. After evaluation of the abstracts and full texts, 11 studies were included. One study was excluded due to suggested overlap [39]. The PRISMA selection flowchart is presented in Fig 1.

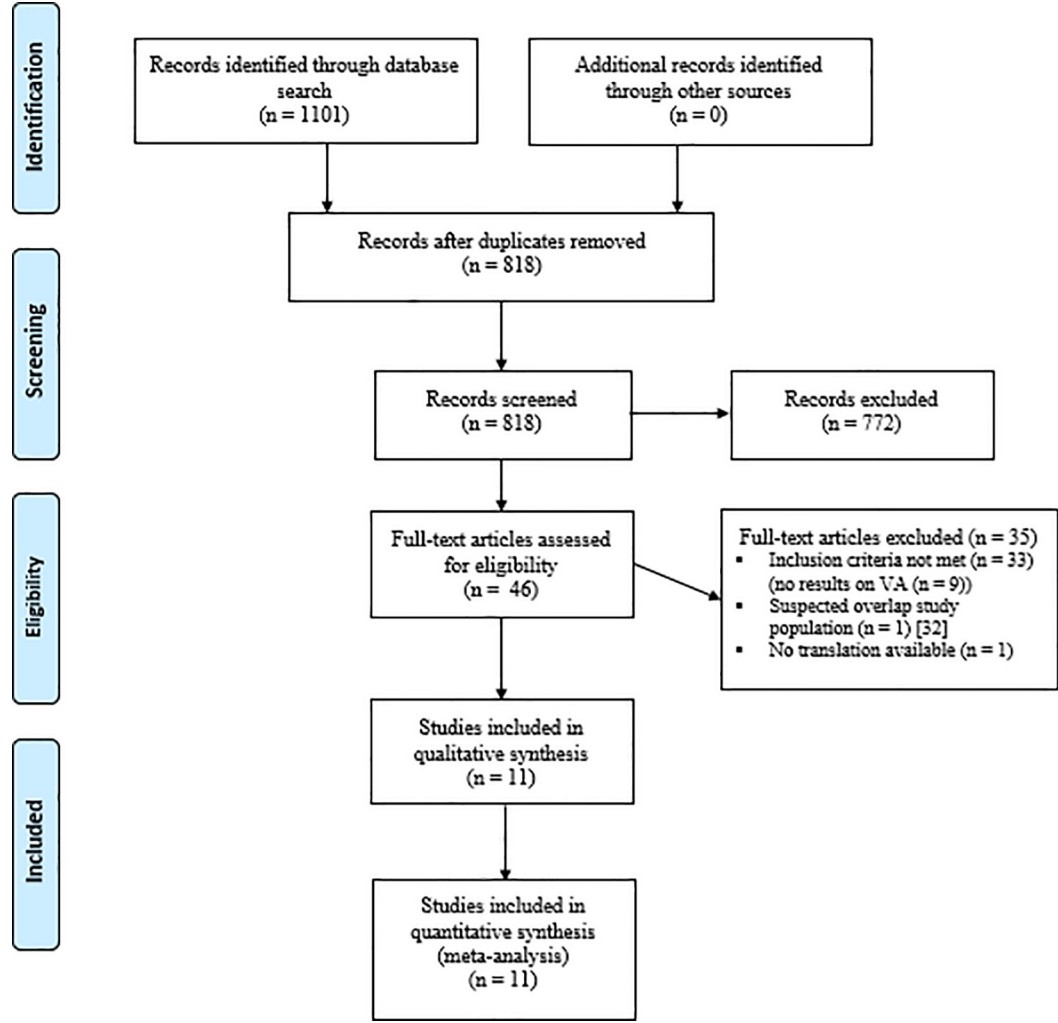

**Fig 1. PRISMA flowchart for identification and selection of studies.**

## Study, patient and treatment characteristics

All 11 included studies were case studies. The studies presented the results of 1,336 patients, of whom 427 had received SAT. Data for the analysis of change in VA were available for 358 patients. Patient characteristics are presented in Table 1. The type of SAT, type of visual test and prognostic factors are shown in Table 2. The cumulative gender distribution among the patients analyzed could be extracted for 178 (of 358) patients from 2 studies [29, 40] (male (N = 88)/female (N = 117)). In 6 studies NF1 status (77%) could only be extracted from the total study population (see Table 1), which also included patients not treated with SAT. The median or mean age at the start of SAT varied from 3.2 to 8 years (range 0.3–17.2 years).

All studies reported on the start of treatment with first-line SAT. They applied various combinations of SAT (see Table 3). SAT regimes were carboplatin-based in 326 of 427 patients (76%). Studies on VEGF or MAPK signaling inhibition did not match the inclusion criteria due to the study volume being < 10 patients or the outcome parameters not matching the focus of this systematic review.

**Table 3. Binocular and/ or monocular change in VA after chemotherapy for OPG.**

| Change in VA/ Author (year) | VA ↑ binoc, N (%) | VA ↔ binoc, N (%) | VA ↓ binoc, N (%) | VA ↑ binoc, N (%) | VA ↔ binoc, N (%) | VA ↓ binoc, N (%) | VA ↑ monoc, N (%) | VA ↔ monoc, N (%) | VA ↓ monoc, N (%) | VA ↑ monoc, N (%) | VA ↔ monoc, N (%) | VA ↓ monoc, N (%) |
|---|---|---|---|---|---|---|---|---|---|---|---|---|
| Massimino et al. (2002) [17] | | | | 10 (45) | 7 (32) | 5 (23) | | | | | | |
| Dalla Via et al. (2007) [41] | | | | 0 (0) | 2 (18) | 9 (82) | | | | 2 (9) | 5 (23) | 15 (68) |
| Massimino et al. (2010) [42] | 7 (41) | 10 (59) | 0 (0) | | | | | | | | | |
| Shofty et al. (2011) [43] | | | | 1 (5) | 4 (21) | 14 (74) | | | | | | |
| Fisher et al. (2012) [30] | 28 (32) | 35 (40) | 25 (28) | | | | 37 (22) | 96 (57) | 35 (21) | | | |
| Kalin-Hadju et al. (2014) [31] | 0 (0) | 11 (69) | 5 (31) | 1 (7) | 7 (50) | 6 (43) | 0 (0) | 27 (79) | 7 (21) | 1 (3) | 24 (71) | 9 (26) |
| Dodgshun et al. (2015) [9] | 5 (14) | 27 (77) | 3 (9) | 4 (11) | 20 (57) | 11 (31) | 7 (10) | 59 (84) | 4 (6) | | | |
| Prada et al. (2015) [44] | | | | 4 (17)* | 6 (27)* | 12 (56)* | | | | | | |
| Doganis et al. (2016) [45] | 7 (44) | 7 (44) | 2 (13) | 4 (25) | 9 (56) | 3 (19) | | | | | | |
| Lassaletta et al. (2016) [18] | | | | 5 (21) | 15 (62) | 4 (17) | | | | | | |
| Falzon et al. (2018) [29] | | | | 19 (21) | 35 (39) | 35 (39) | | | | | | |

## Critical appraisal

All studies were judged as grade 4 evidence according to the CEBM [46]. Critical JBI-CA appraisal of all case series revealed 6 studies with a low risk of bias (see S3 Table).

The focus on change in VA as an outcome parameter was variable among the studies. Four studies presented change in VA as the primary or secondary outcome parameter [9, 29–31]. In other studies change in VA was classed as a higher-order outcome parameter, accompanied mainly by a lack of information on the definition of change in VA.

## Outcome, definition and prognostic factors on change in visual function

After treatment with SAT, 11 included studies (N = 358) showed binocular improvement in VA within the range of 0–45%, stability in the range of 18–77%, and a decrease in VA in the range of 0–82%. Cumulative outcome proportions of the total population are presented in the *Discussion* section. As the studies presented a high variability in the definition of change in VA, we considered these calculations unreliable. This diversity within variables is discussed below.

Change in VF was evaluated in 2 studies [9, 30]. Fisher et al. reported on the outcome of VF in 26 patients: 19% improved, 38% remained stable and 42% decreased. Dodgshun et al. reported 7/35 (20%) abnormalities at diagnosis, of which 2/7 (29%) improved after SAT and in 5/7 (71%) of which VF defects persisted. Neither study reported on the extent of VF loss, the age of the tested population, the type of VF test and the definition of change in VF.

The variability in the definitions of *change in VA* resulted from the various components used. First, different definitions of change in VA were used in each of 4 studies (see Table 1). Fisher et al. applied change ≤ = ≥ 0.2 Snellen lines [30] and second, Falzon et al. ≤ = ≥ 0.2 LogMAR [29]. We consider these definitions as equal, as mostly similar VA cards were used in both studies, which are (partially) validated for conversion to the linear representation of VA: logarithmic minimum angle of resolution (LogMAR), and ≤ = ≥ *0.2 in Snellen lines* is considered equal to ≤ = ≥ 0.2 LogMAR. In these 2 studies the time interval between starting point and evaluation varied greatly (3 months after cessation of SAT [30]–median 6.5 years [29]). Dodgshun et al. defined change in VA according to the ICO (International Council of Ophthalmology: reporting visual loss in research) categories, i.e., a change of 0.3 LogMAR per category [9]. Kalin et al. defined change in VA per category according to the WHO Childhood

Visual Impairment Scale: a change of 0.4 LogMAR per category [31]. Seven studies gave no definition of change in VA, nonetheless reporting on such change.

A second variable regarding change in VA is monocular/binocular evaluation. All studies presented results on binocular change in VA. Four studies presented both binocular and monocular change [9, 30, 31, 41] (see Table 3).

A third variable is term of follow-up. Five studies reported on change in binocular VA from start to within 3 months after the end of SAT [9, 30, 31, 42, 45], 3 of which also provided long-term data [9, 31, 45]. Nine studies published long-term results (range of median follow up was 2.2–8 years after the start of SAT (see Table 1).

Finally, stratification for anatomic location is essential in evaluating change in VA. Only Falzon et al. [29] and Fisher et al. [30] evaluated change in VA per anatomic location; these results are discussed below.

In 6 of these 9 studies tumor progression after first-line SAT was recorded, presenting progression in 86 out of 159 patients (54%). Only in 2 studies was information on change in VA available for between-group analysis on progressed vs. non-progressed OPG, on which we did not perform cumulative analysis as the volumes were too small (18 of 30 progressed) and the studies were not comparable [41, 43].

Prognostic factors for a decrease in VA could not be determined due to the diversity in outcome parameters of the included studies, and therefore they were registered when available per study. Both Falzon et al. [29] and Fisher et al. [30] performed multivariate analyses on prognostic factors for a decrease in VA after CT (see Table 2). Multivariate analysis per subject (with NF1) showed that OPG located posterior to the chiasm ((M)DC stage 3) appears to be a negative risk factor for a decrease in VA [29, 30], but not in nNF1 OPG [29].

Patients under the age of 5 years had a similar negative risk factor for a decrease in VA for NF1 OPG, but not for nNF1 OPG [29].

## Discussion

This systematic review evaluated the impact of SAT for pediatric OPG on VA and VF. Improvement in binocular VA was found in 0–45%, stability in 18–77%, and a decrease in 0–82% of studies after a median follow-up after the start of SAT of 3.7 years (range: cessation of SAT– 8.2 years).

SAT is currently widely applied for progressive pediatric OPG. More than a decade ago, Moreno et al. performed a systematic review on the effect of SAT on visual outcome (1990–2008) [32]. However, all their included studies were of low methodologic quality and were highly heterogeneous. A cumulative decrease in VA after SAT was found in 38% of 174 patients. No analysis was performed on the definitions applied to assess change in VA or VF. Stratification for anatomic location or NF1 status was impossible due to insufficient information. Among the different studies, the following arguments hold true: (1) contamination of treatment results with surgery/radiotherapy and no SAT [47]; (2) a small population of those completing SAT (< 10 patients) [48]; (3) incomplete data on stability of or decrease in VA [49]; and (4) no report available on change in VA [6, 50]. One study was retracted from publication [51].

Our systematic review (search 1990–2020) shows a significant increase in the cumulative population (N = 358) and an increase in studies focusing on change in VA as the primary outcome parameter. The urgency to upgrade future study protocols persists to enable the effect of SAT treatment on visual function to be evaluated, and to enable stratified analysis of NF1 status, age, and anatomic location.

At present in pediatric OPG studies, VA is accepted as the overarching parameter representing visual function [37, 40]. This review showed that no uniform definition of change in

VA was applied in existing studies, as 4 different definitions were used in 4 studies, and in 5 studies no definition was provided. The Response Assessment in Pediatric Neuro-Oncology (RAPNO) working group recommends using the definition of change in VA of $\leq$ or $\geq 0.2$ Log-MAR in future studies on pediatric OPG [37]; this definition is already being applied in the protocols of ongoing studies [52].

At the start of this review, we intended to perform cumulative analysis on the effect of SAT on VA. However, in our opinion, this was not feasible after we had systematically reviewed the 11 included articles, as diverse variables concerning change in VA, such as monocular/binocular evaluation, anatomic location, definition of change in VA/VF and term of follow-up, differed greatly. In working towards an international consensus, we suggest that agreement should first be reached on the definition of change in VA.

A *first variable* contributing to the definition of in change in VA, is the *term of follow–up*. Cumulative analysis on 5 studies on change in VA within 3 months after cessation of SAT shows 27% improvement in VA (N = 47), 52% stability (N = 90) and a 20% decrease (N = 20) in VA. Evaluation of 9 long-term studies (follow-up range of 2.2–8 years) shows improvement in 19% (N = 48), stability in 42% (N = 105) and a decrease in 39% (N = 99). We believe these percentages should be interpreted with caution as definitions of change in VA among the studies differ or are not available.

A *second variable* contributing to the definition of change in VA is the distinction between *monocular or binocular analyses of VA*. Clinical experience suggests that the analysis of 1 or both eyes may differ in the course of OPG, as the anatomic location may result in an asymmetric burden on VA per eye. For example, in the case of a unilateral optic nerve glioma (stage 1 (M)DC), monocular VA may decrease considerably due to the progression of OPG, but as visual function of the other eye is not affected, binocular change in VA can remain unaffected. In this review, the studies defined monocular change in various ways, and therefore this outcome was not comparable. Future assessment of both the monocular and binocular status should evaluate the effect of therapy through per-eye analysis, and should evaluate functional visual disability through 2–eye analysis.

A *third variable* contributing to the definition of change in VA is *anatomic location of OPG*. This requires stratified analysis as the location of (NF1) OPG posterior to the chiasm appears to be a prognostic factor for a decrease in VA after SAT [29, 30], and there appears to be diversity in progression-free survival (PFS) among different anatomic locations [53].

The *fourth variable* contributing to the definition on change in VA is *age at start of treatment*. The combination of ongoing natural development of childhood visual function and known risk factors for the progression of OPG (age (<1 year [53]) or a decrease in VA after SAT (age < 5 year [29, 30]) requires stratification for different age categories. In this review, age at the start of treatment (median 3.2–8 years (range 0.4–17.2 years)) and duration of follow-up after SAT (median 3,7 years (range 0–8,2 years)) varied widely varied among the included studies. Stratification according to age categories was not possible as categorical or individual data were lacking in the majority of the studies.

Optic pathway glioma located in the chiasm and optic tract mostly results in a combination of defects in central (VA) and peripheral vision (VF). In the literature, both VA and VF are considered to mirror each other's function [30]. This assumption could not be substantiated in this review, as only 2 studies (on 33 patients) reported on these parameters. No association between VA and VF could be determined. In addition, a definition of change in VF was lacking. Performing VF tests at a young age (< 7 years) or on children with limited cooperation is highly challenging with a high risk for bias, which could explain the discrepancy between wide integration of VF examination in study protocols and the low level of presentation of results [30].

As, currently, 2D volume changes on MRI are poorly predictive for change in VA [29, 30], other forms of (more objective) examination like optical coherence tomography (OCT) have gained increased attention. OCT has proved to be a potent biomarker for visual loss in the case of screening for (NF1) OPG [54]. Regarding the monitoring of treatment effect, retinal nerve fiber layers (RNFL) appear to be associated with change in visual function. However, larger volume studies on the correlation between change in VA and RNFL or ganglion cell layer–inner plexiform layer are required [54].

Since 2008 treatment options for recurrent pediatric OPG have expanded with VEGF and MAPK pathway inhibitors. In this systematic review of studies on the effect of SAT on change in VA, these treatment modalities were not included as either their outcome parameters had no focus on the effect of visual function (8 studies), or the series were still very small (< 5 patients per series) [22, 55]. Nonetheless, the results on improvement/recovery of VA and/or VF after bevacizumab are considerable and impressive, necessitating future studies of larger volume and quality. Although a thorough and definitive effect analysis of MAPK inhibition on visual function is currently unavailable, several ongoing studies do include such an analysis [52, 56].

The findings of this review should be interpreted in the light of several limitations. First, the included studies presenting an outcome on change in VA are all non-comparable cohort studies with a high level of variability in outcome parameters among them. The rarity of the diagnosis as well as the diversity of location and tumor behavior of OPG make it difficult to perform high-quality prospective studies in this field.

Secondly, we included studies with surgical intervention prior to SAT, which may bias the effect on change in VA. Tumor resection or reducing intracranial pressure by VD can affect visual function and some days to several months may be required to evaluate this effect. In 5 studies, prior to the start of treatment, OPG had been resected or biopsied, or VD had been placed (see Table 2 [9, 18, 29–31]). No information was available on the time interval after surgery, the extension of resection or the surgical effect on VA before the start of SAT. Three studies presented no data on prior surgical therapy [17, 44, 45]. Only Shofty et al. [45] excluded surgery. Surgical intervention frequently needs to be followed promptly by the start of SAT, limiting separate evaluation of the effect of surgery on VA and creating bias regarding the effect of SAT on change in VA.

Thirdly, in this review 6 out of 9 studies covered long-term follow–up, including results on subsequent progression: 54% of OPG cases progressed, of which the majority received sequential therapy, but no cumulative proportion could be calculated due to missing data. Within this time interval different parameters may affect visual function, either positively (e.g., individual potential for visual maturation), or negatively (e.g., a further decrease in VA at progression and during subsequent treatment of OPG).

Fourthly, in our series the incidence of NF1 patients is high (77%), which we consider an unreliable representation. In 6 studies NF1 status was derived from the total study population, including OPG that had received no SAT or other, non-SAT treatment. Several multicenter studies on various treatments for OPG report a lower incidence of NF1 association of between 6 and 27% [6, 57, 58]. One study stated that treatment with SAT renders a higher PFS for NF1-associated OPG compared to nNF1 OPG [2], but this was contradicted in other studies.

In this review stratified analysis on the outcome for NF1/nNF1 was not performed due to the lack of individual data on VA outcome as per NF1 status. The only comparative results on NF1 status were available from Falzon et al.: NF1 is associated with a decrease in VA after SAT when the child is diagnosed with OPG at age ≤ 5 years and the anatomic location is posterior to the chiasm [29].

## Conclusion

This systematic review on the treatment effect of SAT on visual function for pediatric OPG found an improvement in binocular VA in 0–45%, stability in 18–77% and a decrease in 0–82% in 11 studies, including 358 eligible patients. Although in the last decade studies have increased their focus on the effect of SAT on visual function, the quality of the studies persists on level 4 (CEBM) and the high diversity in outcome variables on the definition of *change in VA* limits meta-analysis. Reports on the effect of treatment on VF were scarce and outcome parameters were not defined. Treatment was carboplatin-based in 76% of OPG. No studies reporting on change in visual function after VEGF or MAPK signaling inhibition met the eligibility criteria due to low study volume (N = < 5). Future studies on the effect of SAT for pediatric OPG, including uniform VA monitoring protocols, are needed to evaluate treatment outcomes and to determine prognostic factors for the effect of SAT on visual function.

## Supporting information

**S1 File. Systematic research protocol.**
(PDF)

**S2 File. PRISMA checklist.**
(PDF)

**S1 Table. Search strategy Ovid MEDLINE(R)and epub ahead of print.** Search strategy Ovid MEDLINE(R) and Epub Ahead of Print, In-Process & Other Non-Indexed Citations and Daily 1946 to August 04, 2020. Date of search: 2020-08-06.
(DOCX)

**S2 Table. Search strategy embase classic + embase.** Search strategy Embase Classic+Embase 1947 to 2020 August 05. Date of search: 2020-08-06.
(DOCX)

**S3 Table. JBI-Critical appraisal of included case series.**
(DOCX)

## Author Contributions

**Conceptualization:** Carlien A. M. Bennebroek, Laura. E. Wijninga.

**Data curation:** Laura. E. Wijninga.

**Formal analysis:** Carlien A. M. Bennebroek.

**Investigation:** Jaqueline Limpens.

**Methodology:** Carlien A. M. Bennebroek.

**Project administration:** Carlien A. M. Bennebroek.

**Supervision:** Carlien A. M. Bennebroek.

**Visualization:** Laura. E. Wijninga.

**Writing – original draft:** Carlien A. M. Bennebroek.

**Writing – review & editing:** Jaqueline Limpens, Antoinette Y. N. Schouten-van Meeteren, Peerooz Saeed.

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
