## [Decision Letter · Decision Letter 0]

2 Aug 2021

PONE-D-21-14820

Impact of systemic anticancer therapy in pediatric optic pathway glioma on visual function: a systematic review.

PLOS ONE

Dear Dr. Bennebroek,

Thank you for submitting your manuscript to PLOS ONE. After careful consideration, we feel that it has merit but does not fully meet PLOS ONE’s publication criteria as it currently stands. Therefore, we invite you to submit a revised version of the manuscript that addresses the points raised during the review process.

We look forward to receiving your revised manuscript.

Kind regards,

Jonathan H Sherman

Academic Editor

PLOS ONE

Additional Editor Comments (if provided):

Reviewers' comments:

Reviewer's Responses to Questions

**Comments to the Author**

1. Is the manuscript technically sound, and do the data support the conclusions?

Reviewer #1: Yes

Reviewer #2: Yes

2. Has the statistical analysis been performed appropriately and rigorously? 

Reviewer #1: Yes

Reviewer #2: Yes

3. Have the authors made all data underlying the findings in their manuscript fully available?

Reviewer #1: Yes

Reviewer #2: Yes

4. Is the manuscript presented in an intelligible fashion and written in standard English?

Reviewer #1: Yes

Reviewer #2: No

5. Review Comments to the Author

Reviewer #1: This is a well crafted review of SAT in OPG on visual function. This review highlights the need for prospective studies, as the authors point out, with more uniform parameters. The hope with this type of publication is that future studies will use for uniform methods of VF analysis.

Reviewer #2: I don't think the results are earth-shattering, but it is important to point out that the outcomes regarding visual improvement in OPG chemotherapeutic treatment is inconsistent.

My biggest issue, all due respect, is the English/prose. I would recommend a primary English speaker revise for syntax and grammar prior to publication.

6. PLOS authors have the option to publish the peer review history of their article (what does this mean?). If published, this will include your full peer review and any attached files.

Reviewer #1: No

Reviewer #2: No

---

## [Author Response · Author response to Decision Letter 0]

30 Aug 2021

Response to reviewers:

I herewith send our response to the reviewers received on August 2, 2021, regarding our manuscript entitled ‘Impact of systemic anticancer therapy in pediatric optic pathway glioma on visual function: a systematic review’ (PONE-D-21-14820). 

Concerning point 1, 2 and 3 we have received no comments from the reviewers. We kindly thank you for your review.

4. Is the manuscript presented in an intelligible fashion and written in standard English?

Reviewer #1: Yes

Reviewer #2: No 

Response to reviewer 2: The manuscript has been revised for syntax and grammar by a primary English speaker who is a professional editor. (Editing declaration attached.) 

5. Review Comments to the Author

Reviewer #1: This is a well crafted review of SAT in OPG on visual function. This review highlights the need for prospective studies, as the authors point out, with more uniform parameters. The hope with this type of publication is that future studies will use for uniform methods of VF analysis.

Response to reviewer 1: We thank you for your comments. 

Reviewer #2: I don't think the results are earth-shattering, but it is important to point out that the outcomes regarding visual improvement in OPG chemotherapeutic treatment is inconsistent. 

Response to reviewer 2: The manuscript has been fully revised for syntax and grammar by a primary English speaker (see: OPG and SAT-PLOS ONE-revisited-track changes-25082021) 

The manuscript has been adjusted to meet PLOS ONE's style requirements.

We highly appreciate the suggestions made by the reviewers and we are looking forward to your decision on the suitability of our manuscript for publication in PLOS ONE.

---

## [Decision Letter · Decision Letter 1]

30 Sep 2021

Impact of systemic anticancer therapy in pediatric optic pathway glioma on visual function: a systematic review.

PONE-D-21-14820R1

Dear Dr. Bennebroek,

We’re pleased to inform you that your manuscript has been judged scientifically suitable for publication and will be formally accepted for publication once it meets all outstanding technical requirements.

Kind regards,

Jonathan H Sherman

Academic Editor

PLOS ONE

Additional Editor Comments (optional):

Reviewers' comments:

Reviewer's Responses to Questions

**Comments to the Author**

1. If the authors have adequately addressed your comments raised in a previous round of review and you feel that this manuscript is now acceptable for publication, you may indicate that here to bypass the “Comments to the Author” section, enter your conflict of interest statement in the “Confidential to Editor” section, and submit your "Accept" recommendation.

Reviewer #1: All comments have been addressed

Reviewer #2: All comments have been addressed

2. Is the manuscript technically sound, and do the data support the conclusions?

Reviewer #1: Yes

Reviewer #2: Yes

3. Has the statistical analysis been performed appropriately and rigorously? 

Reviewer #1: Yes

Reviewer #2: Yes

4. Have the authors made all data underlying the findings in their manuscript fully available?

Reviewer #1: Yes

Reviewer #2: Yes

5. Is the manuscript presented in an intelligible fashion and written in standard English?

Reviewer #1: Yes

Reviewer #2: Yes

6. Review Comments to the Author

Reviewer #1: the authors have corrected the prior queries from the reviewers, and have improved the readability of the paper. I would not add any new comments, other than what has been stated in first review. Minor editorial changes make this more readable.

Reviewer #2: Accept without further change. My concerns with syntax and grammar have been addressed. From and neurosurgical perspective I have no concerns or reservations.

7. PLOS authors have the option to publish the peer review history of their article (what does this mean?). If published, this will include your full peer review and any attached files.

Reviewer #1: No

Reviewer #2: No

---

## [Editor Report · Acceptance letter]

7 Oct 2021

PONE-D-21-14820R1 

Impact of systemic anticancer therapy in pediatric optic pathway glioma on visual function: a systematic review 

Dear Dr. Bennebroek:

I'm pleased to inform you that your manuscript has been deemed suitable for publication in PLOS ONE. Congratulations! Your manuscript is now with our production department. 

Kind regards, 

on behalf of

Dr. Jonathan H Sherman 

Academic Editor

PLOS ONE